# Structure and Properties of Electrochemically Synthesized Silver Nanoparticles in Aqueous Solution by High-Resolution Techniques

**DOI:** 10.3390/molecules26175155

**Published:** 2021-08-25

**Authors:** Carla Gasbarri, Maurizio Ronci, Antonio Aceto, Roshan Vasani, Gianluca Iezzi, Tullio Florio, Federica Barbieri, Guido Angelini, Luca Scotti

**Affiliations:** 1Department of Pharmacy, University “G. d’Annunzio” of Chieti-Pescara, Via dei Vestini, 66100 Chieti, Italy; carla.gasbarri@unich.it (C.G.); guido.angelini@unich.it (G.A.); 2Department of Medical, Oral and Biotechnological Sciences, University “G. d’Annunzio” of Chieti-Pescara, Via dei Vestini, 66100 Chieti, Italy; maurizio.ronci@unich.it (M.R.); antonio.aceto@unich.it (A.A.); 3Monash Institute of Pharmaceutical Sciences, Monash, VIC 3800, Australia; roshan.vasani@monash.edu; 4Dipartimento di Ingegneria and Geologia, Università G. d’Annunzio, Via dei Vestini 30, 66013 Chieti, Italy; gianluca.iezzi@unich.it; 5Department of Internal Medicine and Center of Excellence for Biomedical Research (CEBR), University of Genova, 16132 Genova, Italy; tullio.florio@unige.it (T.F.); federica.barbieri@unige.it (F.B.); 6IRCCS Ospedale Policlinico San Martino, 16132 Genova, Italy

**Keywords:** silver nanoparticles, X-ray analysis, electron microscopy, oxidative state, cytotoxicity

## Abstract

The aim of this work was to deeply investigate the structure and properties of electrochemically synthesized silver nanoparticles (AgNPs) through high-resolution techniques such as transmission electron microscopy (TEM), scanning electron microscopy (SEM), Zeta Potential measurements, and matrix-assisted laser desorption/ionization time of flight mass spectrometry (MALDI-TOF-MS). Strong brightness, tendency to generate nanoclusters containing an odd number of atoms, and absence of the free silver ions in solution were observed. The research also highlighted that the chemical and physical properties of the AgNPs seemed to be related to their peculiar oxidative state as suggested by X-ray photoelectron spectroscopy (XPS) and X-ray powder diffraction (XRPD) analyses. Finally, the MTT assay tested the low cytotoxicity of the investigated AgNPs.

## 1. Introduction

Over the last decade, the range of applications of silver nanoparticles (AgNPs) has been continuously developing, given their unusual properties and features. AgNPs have already been successfully employed in many different areas including catalysts, electronic, magnetic, and optical nanomaterials, antibacterial agents, and thermally conductive nanofluids up to their inclusion into textile and cosmetics products [1,2,3,4,5,6,7,8,9]. Recently, strong antibiotic activity of AgNPs against both planktonic and biofilm phenotypes of *Pseudomonas aeruginosa* and other cystic fibrosis-associated bacterial pathogens was also observed [10]. Moreover, the role of these nanoparticles as interactive but not reactive media for azobenzene isomerization has been demonstrated by kinetic, spectroscopic, and Zeta Potential measurements [11].

In general, size, shape, morphology, and physical and chemical properties of metal nanoparticles are strongly affected by the experimental conditions in which their synthesis occurs [12,13].

Different methods have been proposed to produce silver nanoparticles, including laser ablation and the photochemical approach [12,13,14,15]. However, despite its potential versatility both in organic solvents and aqueous solutions [16,17], electrochemical synthesis remains limited to few applications [18,19]. This method is based on different steps: anode oxidation–cathode reduction reactions, hydrogen and oxygen formation from water electrolysis, and side reactions [20]. Only a small amount of hydrogen peroxide is usually needed to reduce the silver oxide in the solution, while stabilizers or coating agents, such as tetrabutyl ammonium salts or poly-(N-vinylpyrrolidone), are added [21,22] in order to control the size and the release of the silver into the surrounding media [23,24].

It is acknowledged that a single surface plasmon resonance band in the UV-Vis region around 400 nm is related to spherical nanoparticles [25], even if some factors can affect the extinction coefficients [26,27]. However, the aggregation of silver nanoparticles may also easily occur in this case, and different techniques can be employed to observe the aggregation phenomena in colloidal systems over time [28,29,30,31,32,33,34,35].

Based on a novel and fast fabrication method using a device built in-house, uncoated and negatively charged AgNPs have been electrochemically synthesized in an aqueous solution and without stabilizer. The obtained AgNPs were spherical and monodispersed in a solution, which showed high stability according to UV–vis spectroscopy analysis, transmission electron microscopy and field emission scanning electron microscopy images, dynamic laser light scattering, and zeta potential measurements [36]. Another application for AgNPs in the photocatalysts field—absorbing light beyond the UV–visible region and catalyzing reactions using the collected upconverted (NIR to UV–visible) photon energy—has received actual attention [37,38]. Moreover, kinetic, spectroscopic, and zeta potential demonstrate the role of these nanoparticles as interactive but not reactive media for azobenzene isomerization.

The aim of this work was to explore the surface and core of the investigated nanoparticles by using high-resolution techniques such as scanning electron microscopy (SEM), transmission electron microscopy (TEM), X-ray photoelectron spectroscopy (XPS), X-ray powder diffraction analysis (XRPD), Zeta Potential measurements, and matrix-assisted laser desorption/ionization time of flight mass spectrometry (MALDI-TOF).

## 2. Materials and Methods

### 2.1. AgNPs Preparation

The modified synthesis of AgNPs in aqueous solution is based on the electrochemical dissolution of pure 99.9% silver electrodes in pure water and applying a low electrical current, as previously described [36]. The power supply and a home-made electronic board were assembled for the synthesis. Voltage, current, reaction time, and total electrical power were optimized and monitored during all the electrochemical processes (Patent Application EP 18181873).

### 2.2. Zeta Potential and Oxidation Reduction Potential (ORP) Measurements

The Zeta Potential analysis was carried out at 25 °C using a 90PLUS BI-MAS (Brookhaven, MS, USA) equipped with digital correlator at a scattering angle of 15°, with a 35 mW He–Ne laser at the wavelength of 660 nm. An electrode ORP (mod 98201, Hanna Instruments) and a dithiothreitol solution (DTT, 20 ug/mL) were employed as standard for ORP analysis.

### 2.3. TEM and SEM Analysis

TEM images were taken after the evaporation of a drop of AgNPs diluted solution on 300 mesh formvar coated nickel grids at 75 kV by using a ZEISS 109 microscope equipped with a Gatan-Orius SC200W-Model 830.10W TEM CCD Camera. SEM images were obtained from the powder after the complete evaporation of the AgNPs solution at 40 °C under a nitrogen flow by using a Sigma 300 Zeiss microscope equipped with an elemental microanalysis apparatus Quantax-200 Bruker.

### 2.4. XPS and XRPD Analysis

XPS analysis was performed using an AXIS Nova spectrometer (Kratos Analytical Inc., Manchester, UK) with a monochromatic Al Kα source at a power of 180 W (15 kV × 12 mA) and a hemispherical analyzer operating in the fixed analyzer transmission mode. XRPD pattern was collected from 4 to 90° of 2θ, with a step scan of 0.02° and 8 s per step with a Rigaku Ultima IV diffractometer. Experimental data were obtained with two different sample states: solid and in solution. The solid phase was obtained by removing the solvent under nitrogen gas. Here only the results obtained with the solid phase sample are reported.

### 2.5. MALDI-TOF Measurements

The samples were prepared by adding 0.5 μL of 0.1% trifluoroacetic acid to 1 μL of 20 ppm AgNPs aqueous solution deposited on the MALDI target. The analysis was performed after drying on an Autoflex Speed TOF/TOF mass spectrometer (Bruker-Daltonics GmbH, Bremen, Germany) equipped with a Smart Beam II 1 kHz laser, in Reflectron Positive (RP) mode, in the range of 20–1.300 Da. The instrument was controlled through Flexcontrol 3.4.135.0 software. The voltages were set as follows: 19.00 kV and 16.70 kV for ion source 1 and 2, respectively, 8.55 kV for the lens, and 21.00 kVand 9.60 kV for reflector 1 and 2, respectively. Pulsed ion extraction time was set to 130 ns. Laser fluence was kept very low to avoid unwanted fragmentation of sample clusters and adducts. The power attenuator was adjusted before the acquisition to maximize resolution, and a laser frequency of 100 Hz was used. For each sample spot, 5000–15,000 total shots were summed.

### 2.6. Cytotoxicity Assay

Cells (2.500 cells/well) were seeded in 96-well plates and treated with AgNPs or vehicle (PBS) in dose-response experiments after 24 h. Cell viability was assessed after 24–48 h of treatment using the MTT assay, which evaluates the mitochondrial function as an index of cell viability by measuring the reduction of 3-(4,5-dimethylthiazol-2-yl)-2,5-diphenyltetrazolium bromide (or MTT assay, Sigma-Aldrich, Milano, Italy) as previously reported [39]. Moreover, the IC_50_ values were calculated using nonlinear regression curve fit analysis with Graph Pad Prism 6.00 (GraphPad Software, San Diego, CA, USA). Data were expressed as the mean ± standard deviation. Cell culture: HEK-293 (human embryonic kidney) cell line was maintained in Dulbecco’s Modified Eagle Medium (DMEM, EuroClone, Lombardy, Italy) containing fetal bovine serum (FBS, Vicenza, Italy) to a final concentration of 10%, 2 mM L-Glutamine (EuroClone, Lombardy, Italy), and 1% penicillin–streptomycin (EuroClone, Lombardy, Italy).

## 3. Results and Discussion

Generally, a high absolute zeta potential value suggests that nanoparticles tend to repulse each other, avoiding any aggregation process [40,41]. The investigated AgNPs showed good stability in an aqueous solution according to the Zeta Potential values in the range −40/−70 mV. Statistical analysis for particle size and distribution performed by TEM on different samples highlighted a high frequency of about 3 nm nanoparticles, as shown in Figure 1.

It is well-known that SEM images provide information about structure, surface, and composition of nanomateria ls and nanoparticles. Examples of SEM images obtained for the AgNPs are reported in Figure 2.

Silver nanoparticles have a nonspherical shape and exhibit a strong “charging-lake”. Generally, charging is related to the lack of conductivity for a specimen in the so-called charging effect [42] due to the accumulation of static electric charge onto the particle surface. The results indicate that the AgNPs were nonconductive and could be considered as an insulating material. Moreover, ORP analysis revealed a value of +400 mV associated with silver nanoclusters in comparison to −90 mV obtained from the DTT standard solution, suggesting oxidative rather than reductive properties of the investigated AgNPs. Furthermore, the absence of free silver ions was demonstrated by ISE analysis [43].

The chemical state of the silver was determined by the position of the peaks associated with the binding energy of the emitted photoelectrons in the XPS spectrum. In particular, the bulk value of silver corresponds to 368.1 eV, and the core level of the binding energy shifts directly provide the metal electronic properties [44]. One of the XPS spectra obtained for the AgNPs is reported in Figure 3 as an example.

The peaks at 374.53 and 373.32 eV are associated with metallic and oxidized Ag 3d_3/2_, respectively, while the peaks at about 368.53 and 367.32 eV are associated with oxidized and metallic Ag 3d_5/2_ [45]. The binding energy of metallic silver is higher than the binding energy observed in oxidized silver, therefore, the former represents the dominant state for the investigated AgNPs [46]. Moreover, the presence of Ag^III^ was pointed out by the shift of the values toward higher binding energies, in agreement with previous studies [47,48].

X-ray powder diffraction analysis was performed to evaluate the oxidative state of the AgNPs. The XRPD pattern can be observed in Figure 4. The pattern peaks at 2θ values of 38.10, 44.28, 64.42, 77.40, and 81.50 can be attributed to the reflection of (111), (200), (220), (311), and (222) planes of the face-centered cubic silver, respectively [49,50].

The diffraction profiles of as-prepared AgNPs were broadened compared to those of bulk silver, revealing the formation of silver nanoparticles [51].

The other labeled diffraction peaks may be attributed to the Ag_3_O_4_ and AgO crystalline phases on the surface of the AgNPs [49,52].

Silver nanoclusters can be also detected by MALDI-TOF mass spectroscopy analysis. Interestingly, our results showed aggregates in the low mass region that were mainly composed of an odd number of atoms (Table 1).

Finally, the cytotoxicity of the investigated AgNPs was tested by an MTT assay on HEK-293 (human embryonic kidney). The dose-dependent curve, shown in Figure 5, demonstrated lower toxicity of AgNPs in comparison to that of AgNO_3_.

## 4. Conclusions

AgNPs synthesized electrochemically without adding any stabilizers or coating agents are negatively charged and monodisperse in an aqueous solution. A large population of about 3 nm in size and strong brightness was observed. Metallic rather than oxidized silver was the dominant state for the investigated AgNPs as suggested by an X-ray photoelectron spectrum, while the presence of oxides and the tendency to form nanoclusters was demonstrated by XPRD analysis and MALDI-TOF measurements (see Appendix A). Finally, the cytotoxic effect observed by the MTT assay confirmed that the investigated AgNPs could improve the use of nanoparticles in aqueous media for a wide range of applications, both in chemical and biological fields.

## Figures and Tables

**Figure 1 molecules-26-05155-f001:**
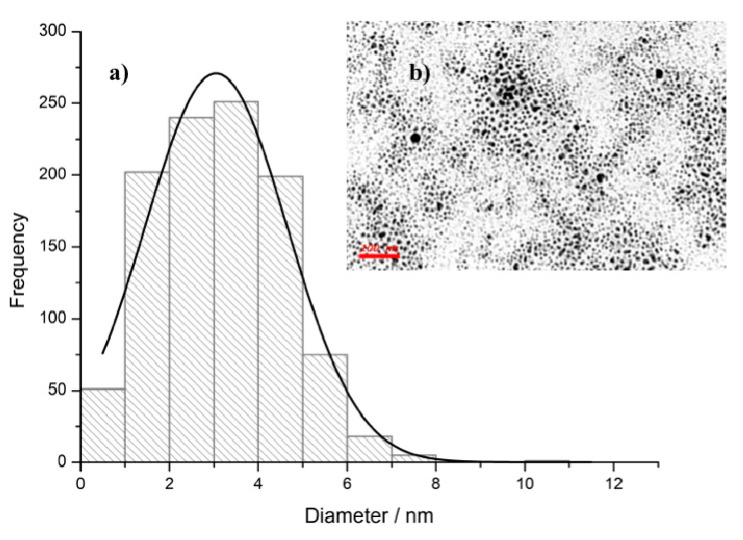
Panel (**a**): particle size distribution histogram indicating high frequency of nanoparticles in the 1–5 nm range and lack of nanoparticles larger than 8 nm. In panel (**b**), a TEM micrograph of the investigated AgNPs is shown (scale bar: 100 nm). The statistical distribution of nanoparticles was obtained by the elaboration of TEM images with the ImageJ program and calculated with StatPlus-2 or Origin software.

**Figure 2 molecules-26-05155-f002:**
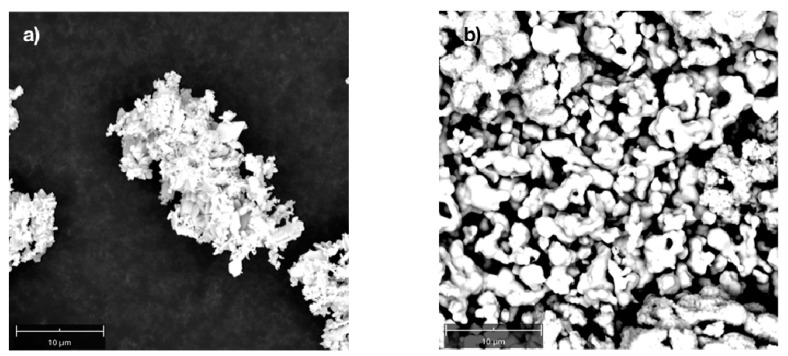
Example of an SEM image of the investigated AgNPs in a solid sample. Evidence of a typical charge effect of nonconductive samples (specimen) that were not treated with a “coated technique”. (**a**) The figure was obtained with the drop technique on a metal sample holder (specimen) covered with charcoal. (**b**) The figure represents the magnification useful for the elementary analysis.

**Figure 3 molecules-26-05155-f003:**
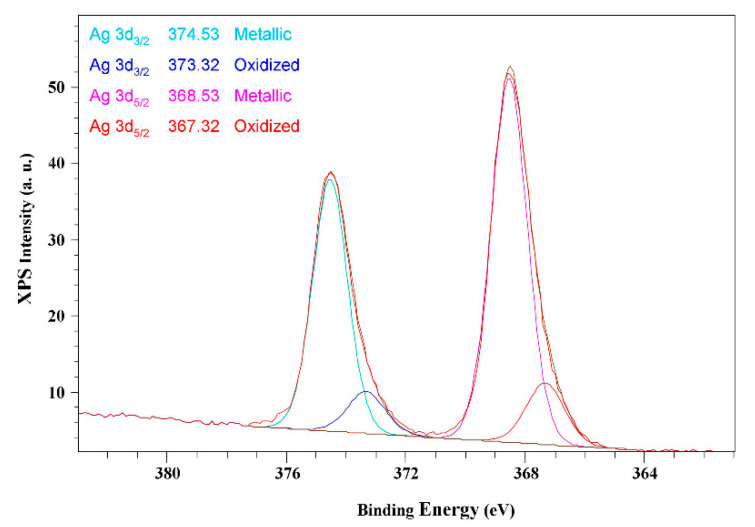
X-ray photoelectron spectrum of AgNPs. The binding energies are referred to as Ag 3d_3/2_ and Ag 3d_5/2_ in metallic and oxidized states.

**Figure 4 molecules-26-05155-f004:**
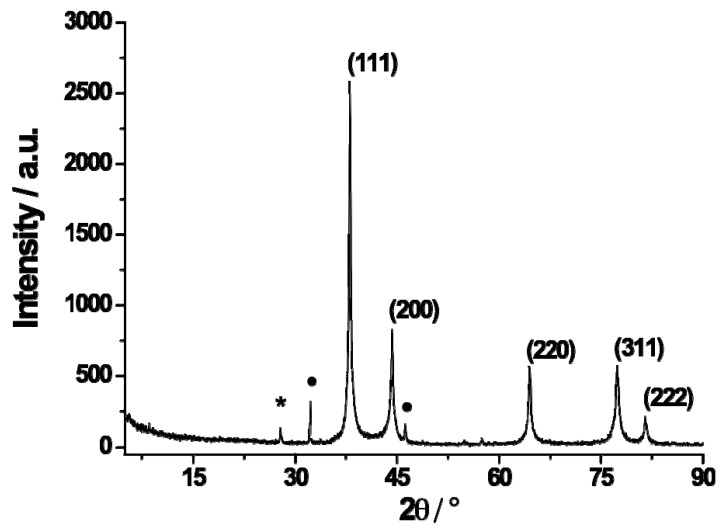
The XRPD pattern for the investigated AgNPs of Ag (0). The peaks marked with * and • are due to Ag_3_O_4_ and AgO, respectively.

**Figure 5 molecules-26-05155-f005:**
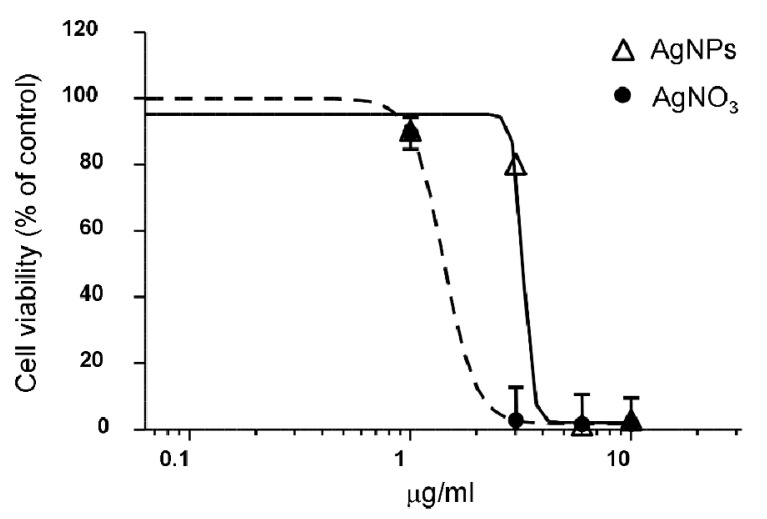
Dose-response curves of the effects of AgNPs and AgNO_3_ on HEK-293 viability, evaluated by an MTT assay after 48 h of treatment. Data represent the mean ± SEM of independent experiments, performed in quadruplicate.

**Table 1 molecules-26-05155-t001:** Detectable (+) and not detectable (−) Ag species by MALDI-TOF mass spectrometry.

Ag Species Detection	Ag	Ag_2_	Ag_3_	Ag_4_	Ag_5_	Ag_6_	Ag_7_	Ag_8_	Ag_9_	Ag_10_	Ag_11_	Ag_12_
Maldi-TOF MS Analysis	+	+	+	−	+	−	+	−	+	−	+	−

## Data Availability

https://materialsproject.org/materials/mp-1605/ (accessed on 14 July 2021).

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
