# Peer review of "Structure and Properties of Electrochemically Synthesized Silver Nanoparticles in Aqueous Solution by High-Resolution Techniques"

_molecules, 2021, doi:10.3390/molecules26175155_

Round 1

Reviewer 1 Report

Authors addressed all concerns. 

Author Response

Reviewer 1

We thank the referee for positive observations.

Reviewer 2 Report

Dear authors, the paper "Structure and properties of electrochemically synthesized silver nanoparticles in aqueous solution by high resolution techniques".

Although the theme is interesting to the authors, the way it is presented is not clear.

For example: It was an aqueous solution monodisperse, but there is no evidence of monodisperse in figure 1. The result shows that the nanoparticles are agglomerated. There is no scale in the nanoparticle figure.

The text is poorly presented with few results that evidence the writing.

Author Response

Reviewer 2

Dear authors, the paper "Structure and properties of electrochemically synthesized silver nanoparticles in aqueous solution by high resolution techniques".

Although the theme is interesting to the authors, the way it is presented is not clear.

For example: It was an aqueous solution monodisperse, but there is no evidence of monodisperse in figure 1. The result shows that the nanoparticles are agglomerated. There is no scale in the nanoparticle figure.

The text is poorly presented with few results that evidence the writing

Answers are in red color.

The mono-dispersion evidence is not reported in the present manuscript but is detailed reported in ref [32] of the bibliography. However, only for referee information, we report the DLS polydispersity analysis (multimodal and lognormal)

If deemed appropriate the Editor could upload the figure as supplementary materials.

“The result shows that the nanoparticles are agglomerated.” 

The results presented in Fig.1 and 2 are obtained not in solution but in solid agglomerated state due to the drop technique on a metal sample holder (specimen), in which the solvent was removed under vacuum and by nitrogen gas.

“There is no scale in the nanoparticle figure.”

The scale is reported in the figure caption and now the reference scale is also present in the figure.

“The text is poorly presented with few results that evidence the writing”

The presentation of new version has been improved. The principal chemico-physical characteristics have been determined and these explain the functional properties of our silver formulation.

Reviewer 3 Report

The authors describe the properties of electrochemically produced silver nanoparticles (AgNPs) by using different techniques and their cytotoxic effect determined by the MTT assay.

The main result is that the AgNPs in solution consists mainly of metallic silver with some contribution of silver oxide content. The MTT assay shows that the AgNPs are cytotoxic for HEK-293 cells, however less compared to the AgNO3 solution. This is not surprising because the cytotoxic effect results mainly from silver ion species and the AgNPs consists mainly of metallic silver.

The Reviewer has found the following critical points:

  1. There are number of sentences or paragraphs where the text is highlighted in yellow – why? It seems to the Reviewer that the manuscript is not a final version.
  2. Lines 46-49: Sentence not clear – doubling of words.
  3. Line 63: “may also easily occur in this case and different techniques can be employed to observe” ; the word “and” is missing.
  4. Line 126: “laser fluence” and not “laser fluency”
  5. Line 151: Figure 1 (“1” is missing)
  6. Figure 1: Panel a and Panel b are not indicated in the Figure
  7. Line 181, Figure 2: There is no explanation what the difference between the left and the right image is. The Reviewer supposes that the right image results from a measurement with higher magnification.
  8. Line 232: The table 1 is too large.
  9. Line 246: “in aqueous solution” and not “an aqueous solution”
  10. Lines 360-364: References 53 and 54: why highlighted with yellow?
  11. The Reviewer wonders whether the data presented in the manuscript are unique – meaning not published elsewhere. However, one can easily find that the data are already given to the readers in researchgate.net. The present submitted manuscript is slightly rewritten in the text – the data and figures are however exactly the same. Can the authors comment on this – see the link: (7) (PDF) Structure and Properties of Electrochemically Synthesized Silver Nanoparticles in Aqueous Solution by High Resolution Techniques (researchgate.net)

Author Response

Reviewer 3

Answers are in red color.

“The authors describe the properties of electrochemically produced silver nanoparticles (AgNPs) by using different techniques and their cytotoxic effect determined by the MTT assay.

The main result is that the AgNPs in solution consists mainly of metallic silver with some contribution of silver oxide content. The MTT assay shows that the AgNPs are cytotoxic for HEK-293 cells, however less compared to the AgNO3 solution. This is not surprising because the cytotoxic effect results mainly from silver ion species and the AgNPs consists mainly of metallic silver.”

The main result of our AgNPs formulation is that although it is bactericidal, at the same concentration however less toxic toward eukaryotic cells.

The Reviewer has found the following critical points:

  1. 1.There are number of sentences or paragraphs where the text is highlighted in yellow – why? It seems to the Reviewer that the manuscript is not a final version.

The yellow color which highlights sentences and paragraph , has been removed.

  1. 1.Lines 46-49: Sentence not clear – doubling of words.

The sentence has been corrected.

  1. 1.Line 63: “may also easily occur in this case and different techniques can be employed to observe” ; the word “and” is missing.

The sentence has been corrected.

  1. 1.Line 126: “laser fluence” and not “laser fluency”

The sentence has been corrected.

  1. 1.Line 151: Figure 1 (“1” is missing)

The sentence has been corrected.

  1. 1.Figure 1: Panel a and Panel b are not indicated in the Figure

The sentence has been corrected.

  1. 1.Line 181, Figure 2: There is no explanation what the difference between the left and the right image is. The Reviewer supposes that the right image results from a measurement with higher magnification.

In Fig 2 the caption text has been improved.

  1. 1.Line 232: The table 1 is too large.

The table 1 now is smaller.

  1. 1.Line 246: “in aqueous solution” and not “an aqueous solution”

The text has been corrected.

  1. 1.Lines 360-364: References 53 and 54: why highlighted with yellow?

The yellow color which highlights sentences and paragraph , has been removed.

  1. 1.The Reviewer wonders whether the data presented in the manuscript are unique – meaning not published elsewhere. However, one can easily find that the data are already given to the readers in researchgate.net. The present submitted manuscript is slightly rewritten in the text – the data and figures are however exactly the same. Can the authors comment on this – see the link: (7) (PDF) Structure and Properties of Electrochemically Synthesized Silver Nanoparticles in Aqueous Solution by High Resolution Techniques (researchgate.net)

I and other coauthors thank very much the reviewer for this information. No co-author was informed and therefore everyone was surprised that a non-recent version of the manuscript was sent to the social network (Researchgate) before the final submission to the "Molecules" journal even if in a preprint submission version . We do not agree with this autonomous initiative of a single researcher, just as we do not take any responsibility for any future possible consequences. As Principal Investigator and corresponding author I confirm that the work has not never been submitted to any other scientific journal.

Round 2

Reviewer 2 Report

The authors address all the points suggested in this new version and the paper  entitled "Structure and properties of electrochemically synthesized silver nanoparticles in aqueous solution by high resolution techniques" is ready to accept.

Author Response

We thank reviewer 2 for the positive report form.

Reviewer 3 Report

Line 150: A point after “1” is missing → “Figure 1.”

Figure 1: The Reviewer cannot see a difference of the legend text in Fig. 1 compared to the original manuscript. “a” und “b” should be indicated in the images themself. Although it seems to be obvious what is “panel a” and what is “panel b” – nevertheless, the Figure should be clearly indicated.

Figure 2: In the legend of Fig. 2 there needs a clear assignment: Either one uses a) and b) - then a) and b) should be written on the image itself or one uses “left” respective “right” without indicating the images themselves with a) and b).

Author Response

Line 150: A point after “1” is missing → “Figure 1.”

In line 150 after 1 now there is a point.

Figure 1: The Reviewer cannot see a difference of the legend text in Fig. 1 compared to the original manuscript. “a” und “b” should be indicated in the images themself. Although it seems to be obvious what is “panel a” and what is “panel b” – nevertheless, the Figure should be clearly indicated.

In figure 1 now “a)” e “b)” has been inserted in the images themself.

Figure 2: In the legend of Fig. 2 there needs a clear assignment: Either one uses a) and b) - then a) and b) should be written on the image itself or one uses “left” respective “right” without indicating the images themselves with a) and b).

In figure 2 now “a)” e “b)” has been inserted in the images themself.

This manuscript is a resubmission of an earlier submission. The following is a list of the peer review reports and author responses from that submission.

Round 1

Reviewer 1 Report

Authors reported electrochemically synthesized AgNPs. This work is interesting. However, the author needs to improve the quality of the manuscript.

The author needs to check the entire manuscript for spelling errors or formatting errors... For examples in abstract, ‘nighlighted’………..

Give full abbreviation at first time use in the text.. for example … ‘MTT’ ‘ORP’

Authors need to describe clearly the synthesis procedure in this manuscript even it is reported elsewhere…

Why does the SEM image have the AgNP particle size in micrometers? Please check

In introduction, authors should highlights the importance of AgNPs and their compounds in different applications and please add these relevant articles in revision: (i) Chemical Engineering Journal 421, 129687; (ii) Journal of Cluster Science 32 (3), 711-718;

In XPS, the author needs to check whether oxygen reacts with Ag. Therefore, please provide the O 1s spectrum of AgNPs. However, in XRD is confirmed.

Reviewer 2 Report

Review of molecules-1283815, “Structure and properties of electrochemically synthesized silver nanoparticles in aqueous solution by high resolution techniques”, by C. Gasbarri, M. Ronci, A. Aceto, et al.

My qualification for giving this manuscript a serious review rests on the fact that I do research in this area. My conclusion is that the authors use concepts and techniques of which they are unfamiliar. As a result, many of their interpretations are simply wrong. I suggest that the manuscript be rejected. My reasons follow:

  • Line 61: the nanoparticles in Figure 2 are hardly spherical and monodisperse.
  • Line 76: the method of preparation is not sufficiently described.
  • Line 96: the XPS method is not sufficiently detailed.
  • Line 135: in Figure 1, panel b, what am I looking at?
  • Line 144: the figure caption is not sufficiently detailed.
  • Line 145: the authors use the term, “brightness”. I have no idea what this means or how it indicates a lack of conductivity. If they mean “sheen” or “reflectivity”, those terms come from the electron sea, and represent conductivity. My own results on similarly sized Ag nanoparticles indicate them to be conductive. Besides, if they were insulating, the XPS peaks would become distorted.
  • Line 161: the XPS interpretations are simply wrong. The 3d5/2 peak at 367.3 eV is due to zerovalent Ag, and that at 368.5 eV, to oxidized Ag, not the reverse. The C1s and O1s spectra are absent, which makes it impossible to see how the energy was calibrated, how many oxide peaks there are, and which are due to oxidized C or Ag.
  • The authors should familiarize themselves with a more up-to-date view of Ag oxides. There are only two: AgI2O and AgIAgIIIO2.
  • Line 171: the reason the XRD peaks are broadened is that the crystals have less perfection.
  • Line 174: the unknown peaks in Figure 4 may be identified using one of the powder diffraction databases available on the Internet.
  • Line 183: cytotoxicity conclusions are based on one non-zero value for each plot in Figure 5? Really?

In addition:

  • Lines 37 and 39: references 37 and 38 are out of order.
  • The grammar is poor, suffering from misspellings, typing mistakes, unnecessary mid-sentence capitalization and unsuitable words.
  • Abbreviations are not defined.
  • Line 183: I would be loath to base a cytotoxicity conclusion on one type of test.

Based on all this, I reiterate my suggestion of rejection.